# Prognostic Implications of Intratumoral Budding in Colorectal Cancer: Detailed Analysis Based on Tumor-Infiltrating Lymphocytes

**DOI:** 10.3390/jcm13010134

**Published:** 2023-12-26

**Authors:** Jung-Soo Pyo, Ji Eun Choi, Nae Yu Kim, Kyueng-Whan Min, Dong-Wook Kang

**Affiliations:** 1Department of Pathology, Uijeongbu Eulji Medical Center, Eulji University School of Medicine, Uijeongbu-si 11759, Republic of Korea; jspyo@eulji.ac.kr (J.-S.P.); kyueng@eulji.ac.kr (K.-W.M.); 2Department of Pathology, Chungnam National University Sejong Hospital, 20 Bodeum 7-ro, Sejong 30099, Republic of Korea; b612elf@cnuh.co.kr; 3Department of Internal Medicine, Uijeongbu Eulji Medical Center, Eulji University School of Medicine, Uijeongbu-si 11759, Republic of Korea; naeyu46@eulji.ac.kr; 4Department of Pathology, Chungnam National University School of Medicine, 266 Munhwa Street, Daejeon 35015, Republic of Korea

**Keywords:** colorectal cancer, intratumoral budding, tumor-infiltrating lymphocyte, immunoscore, prognosis

## Abstract

Background: This study aims to understand the clinical and pathological importance of intratumoral budding (ITB) in colorectal cancer (CRC) and its relationship with tumor-infiltrating lymphocytes (TILs). CRCs can be classified into hot (high immunoscore (IS)) and cold (low IS) tumors. Methods: We investigated the number of ITBs in a hotspot area and categorized them into high-ITB (≥5) and low-ITB (<5) groups. The clinicopathological significance of ITB in human CRCs was evaluated, and a detailed analysis based on tumor-infiltrating lymphocytes (TILs) was also performed. Results: High ITB was identified in 59 of 266 CRC cases (22.2%). High ITB significantly correlated with a poorly differentiated tumor, lympho-vascular invasion, perineural invasion, higher pT stage, lymph node metastasis, and higher metastatic lymph node ratio. High ITB was also significantly correlated with a low IS and low CD8-positive lymphocytic infiltrate. The number of ITBs was substantially higher in the low-IS group than in the high-IS group (3.28 ± 3.31 vs. 2.19 ± 2.59; *p* = 0.005). High ITB significantly correlated with worse overall survival (*p* = 0.004). In the low-IS group, CRCs with high ITB had a significantly worse prognosis than those with low ITB (*p* = 0.021). However, there was no significant difference in prognosis between the high- and low-ITB groups in the high-IS group (*p* = 0.498). Conclusions: Taken together, high ITB was significantly correlated with aggressive tumor behaviors and worse survival in patients with CRCs. In addition, ITB can be useful for the prognostic stratification of CRCs with low IS.

## 1. Introduction

Colorectal cancers (CRCs) are affected by various tumor microenvironments. Tumor budding (TB) is a representative feature in the histology. The definition of TB is single cells or clusters of tumor cells. A cluster of tumor cells has no more than four tumor cells [1]. TB encompasses both intratumoral TB (ITB) and peritumoral TB (PTB). PTB, evaluated at the invasive front, is graded based on the number of PTBs in a hotspot and can only be assessed in endoscopic or surgically resected CRCs. In contrast, ITB refers to TB within the main tumor body and can be evaluated even in biopsy specimens. The International Tumor Budding Consensus Conference (ITBCC) recommended describing PTB at invasive fronts but not ITB [1]. In the previous meta-analysis, the estimated rate of ITB was 23.3% in CRCs from 13 eligible studies [2]. This rate is considered to be common in daily practice. ITB has been associated with lymph node metastasis [1,3,4,5]; however, the ITBCC considers the evidence grade for this association to be low [1]. The clinicopathological significance of ITB, including its association with lymph node metastasis and variation according to tumor differentiation in CRCs, has been previously reported [6]. These features were confirmed through a meta-analysis [2]. In the previous meta-analysis, ITB was significantly correlated with tumor grading, histologic subtype, lymphatic invasion, perineural invasion, pT stage, and lymph node metastasis [2]. The rate of ITB was significantly higher in CRCs with lymph node metastasis than in those without [7,8,9,10]. CRCs with higher pT stage (pT3–4) had higher rates of ITB than those with a lower pT stage (pT1–2) [2]. In addition, there were significant correlations between ITB and disease-free and overall survival [2]. A previous meta-analysis has confirmed the clinicopathologic significance of ITB [2]. However, more detailed studies are needed, as the clinicopathological importance of ITB may change depending on the method of evaluation.

ITB is significantly correlated with tumor regression grade after neoadjuvant chemoradiotherapy in rectal cancer patients [4]. Rectal cancers showing near-complete or complete response (tumor regression grade 1) exhibited no TB in preoperative biopsy specimens [4]. Giger et al. reported that high ITB in preoperative biopsied specimens of CRCs, including both colon and rectum, was significantly correlated with lymph node and systemic metastases [3]. In these studies, only ITB in preoperative biopsy specimens was evaluated. In addition, PTB measurements can be limited by the depth of the tumor. In daily practice, the evaluation of PTB is difficult in CRCs with submucosal invasion. Although ITB is not included in existing guidelines, understanding its significance is important, as ITB can be evaluated in preoperative biopsied specimens.

Malignant tumors, including CRC, are affected by the tumor microenvironment [11]. The importance of tumor-infiltrating lymphocytes (TILs) is growing in the application of immunotherapy [12]. In addition, the prognostic implications of TILs have been reported in CRC [12,13]. Immunoscore (IS) is known as a useful parameter for evaluating TILs. Immunohistochemistry for CD3 and CD8 is applied in evaluating IS. According to the definition of IS, scores of TILs through CD3 and CD8 immunohistochemistry are obtained from two regions: the core and invasive margin of the tumor [12]. The attempts have been to determine the prognostic implications according to the types of immune cells [12]. In addition to CD3- and CD8-positive T lymphocytes, a variety of other immune cells, including M1 macrophages and natural killer cells, can be affected. The relationship between ITBs and TILs was not fully understood. Of course, the association with CD3- and CD8-positive T-lymphocytes was also unclear. In the previous meta-analysis, the correlation between ITB and TILs was evaluated, and it was not found to be significant [2].

The present study aimed to elucidate the clinicopathological and prognostic implications of ITB in CRCs using human CRC tissues. Considering the potential importance of the interaction with tumor-infiltrating lymphocytes (TILs), we investigated the correlation between ITB and TILs in CRCs. Furthermore, due to the heterogeneity of the intermediate grade of ITB [1], we compared 2- and 3-tier grading systems to evaluate the grading system’s usefulness. Prognostic stratification was also attempted based on ITB and immunoscore (IS) grouping.

## 2. Materials and Methods

### 2.1. Patients and Specimens

The medical records of 266 patients who underwent surgical resection for CRCs at Eulji University Medical Center, Eulji University School of Medicine (Republic of Korea), from 1 January 2001, to 31 December 2010, were retrospectively analyzed. We examined the number of ITBs in a hotspot area using a microscope at 200× magnification on hematoxylin and eosin (H&E)-stained glass slides. Additionally, medical charts, pathological records, and glass slides were reviewed to obtain information on clinicopathological characteristics. The parameters investigated in this study included age, sex, tumor size, location, differentiation, tumor depth, lympho-vascular invasion, perineural invasion, lymph node metastasis, metastatic lymph node ratio, distant metastasis, and pathologic tumor–node–metastasis (pTNM) stages. All histological features were evaluated by the 8th edition of the AJCC Cancer Staging Manual [14]. The tumor location was divided into the right and left colon. Rectal cancers were included in the left colon. The metastatic lymph node ratio is defined as the ratio of the number of metastatic lymph nodes to the number of examined lymph nodes. We evaluated the pathologic tumor (pT) stage by dividing it into two groups: pT1–2 and pT3–4. The pTNM stage was divided into two groups: pTNM I–II and III–IV groups. This protocol was reviewed and approved by the Institutional Review Board of Uijeongbu Eulji University Hospital (approval number: UEMC 2023-08-008).

### 2.2. Evaluation of Intratumoral Budding and Immunoscore

In this study, the number of ITBs was evaluated at a hotspot within the tumor. The area examined for ITB was 0.785 mm^2^ under the microscope. The ITBs were categorized into high and low groups, with the division based on a threshold of five ITBs. All immunohistochemically stained slides for CD3 (Leica Biosystems, Newcastle Upon Tyne, UK) and CD8 (Leica Biosystems) were scanned using a Pannoramic MIDI II (3DHISTECH, Budapest, Hungary). Images of two regions: the core of the tumor and the invasive margin, were captured using CaseViewer 2.0 (3DHISTECH). Using NIH Image Analysis software (version 1.6.0, National Institute of Health, Bethesda, MD, USA), CD3- and CD8-immunoreactive lymphocytes were quantified in these images, with a consistent intensity threshold set for analysis. CD3- and CD8-immunoreactive lymphocytes were expressed as pixel counts in each region. In the present study, the cut-off for categorizing patients was set at the median value of these pixel counts. Based on this cut-off, patients were classified into two groups: high (score 1) and low (score 0). The IS was defined as the sum of the scores from both regions and was categorized into high (IS 3–4) and low (IS 0–2) groups.

### 2.3. Statistical Analysis

Statistical analyses were conducted using SPSS software, version 22.0 (SPSS, Chicago, IL, USA). To determine the significance of the correlation between ITB and clinicopathological characteristics, including sex, tumor size, location of tumor, tumor differentiation, vascular, lymphatic, and perineural invasions, pathologic tumor (pT) stage, lymph node metastasis, and distant metastasis, we used the χ^2^ test or the Fisher exact test (two-sided). Comparisons of ITB with age, tumor size, and metastatic lymph node ratio were performed using the two-tailed Student’s *t*-test. The correlations between ITB, IS, and TILs were also analyzed using the χ^2^ test. Additionally, the variation in ITB numbers across different IS groups was assessed using the two-tailed Student’s t-test. Survival curves were estimated using the Kaplan–Meier product limit method, and differences between these curves were evaluated using the log-rank test. The results were considered statistically significant at *p* < 0.05.

## 3. Results

### 3.1. The Clinicopathological Significances of Intratumoral Budding

Representative images of ITB and TILs are shown in Figure 1A–D. In addition, representative images for the immunohistochemistry of CD3 and CD8 are shown in Figure 1E,F. High ITB was observed in 59 out of 266 CRCs, accounting for 22.2%, as detailed in Table 1. CRCs with high ITB were significantly associated with poor differentiation (*p* = 0.045). Additionally, high ITB was significantly associated with frequent vascular, lymphatic, and perineural invasion, as well as lymph node metastasis (*p* = 0.035, *p* < 0.001, *p* < 0.001, and *p* = 0.026, respectively). In high ITB cases, the rates of vascular, lymphatic, and perineural invasions were 16.9%, 50.8%, and 35.6%, respectively. However, in low ITB cases, the rates of vascular, lymphatic, and perineural invasions were 6.8%, 19.3%, and 11.1%, respectively. Furthermore, significant correlations were observed between high ITB and a higher pT stage, as well as with an increased metastatic lymph node ratio (*p* = 0.041 and *p* = 0.008, respectively). The metastatic lymph node ratios of high- and low-ITB groups were 0.21 (standard deviation [SD] 0.27) and 0.11 (SD 0.20), respectively. While high ITB was correlated with distant metastasis and higher pTNM stages, these differences were not statistically significant (*p* = 0.055 and *p* = 0.055, respectively). Tumor sizes were 5.04 cm (SD 1.82 cm) and 5.58 cm (SD 2.13 cm) in the high- and low-ITB groups, respectively. However, there was no statistical difference in tumor size between the high- and low-ITB groups (*p* = 0.074). There were no significant differences between ITB and other clinicopathological features, such as age, sex, and tumor location.

### 3.2. The Correlation between Intratumoral Budding and Survival

Next, the correlation between ITB and overall survival was evaluated in CRCs. Patients with high ITB had worse overall survival than those with low ITB (*p* = 0.004; Figure 2A). Additionally, to compare the criteria, the prognostic significance of ITB was evaluated using a 3-tier system. In the analysis using a 3-tier system, both high (≥10 buds) and intermediate (5–9 buds) ITB grades were significantly associated with a worse prognosis compared to the low grade (overall *p* = 0.009; Figure 2B). However, there was no significant difference between high and intermediate ITB grades (*p* = 0.340).

### 3.3. The Correlation between Intratumoral Budding and Tumor-Infiltrating Lymphocytes

In the present study, TILs were evaluated through the classification of IS. The number of ITBs was significantly higher in cases with low IS compared to those with high IS in CRCs (3.28 ± 3.31 vs. 2.19 ± 2.59, *p* = 0.005; Figure 3). A strong correlation was found between high ITB and low IS (*p* < 0.001; Table 2). CRCs with high IS showing a higher rate of low ITB than those with low IS. Upon detailed analysis, CD8-positive lymphocytes were significantly more prevalent in the high-ITB group than in the low-ITB group (*p* = 0.044), whereas this was not the case for CD3-positive lymphocytes (*p* = 0.283). High ITB was found in 17 of 93 CRCs with high levels of CD3-positive lymphocytes (18.3%). In CRCs with low levels of CD3-positive lymphocytes, high ITB was found in 42 of 173 CRCs (24.3%). In patients with low IS, a significant correlation was observed between high ITB and worse overall survival (*p* = 0.021; Figure 4A). However, in the high IS group, there was no significant difference in overall survival between patients with high and low ITB (*p* = 0.498; Figure 4B).

## 4. Discussion

TB can be observed both within the tumor and at the invasive front. According to the ITBCC guidelines, PTB is specifically investigated at the hotspot of the invasive front [1]. However, these guidelines do not recommend evaluating ITB for assessing TB [1]. To the best of our knowledge, this study is the first to attempt to elucidate the correlation between ITB and TILs in CRC. Additionally, the prognostic implications of ITB were evaluated and stratified using TILs.

Proper evaluation of ITB is crucial for understanding its correlation with both biopsied and surgical specimens. In preoperatively biopsied specimens, TB is indicative of ITB rather than PTB. ITB may be more relevant than PTB in determining its association with the tumor environment. While PTB is suitable for investigating the microenvironment at the invasive front, it may not fully represent the intratumoral microenvironment. If TB is related to the epithelial–mesenchymal transition into the surrounding stroma, understanding its relevance to the intratumoral microenvironment becomes essential. Various factors related to the tumor microenvironment can influence tumor growth and invasiveness. These factors may also affect ITB. Previous studies have evaluated differences in clinicopathologic features of CRC based on the presence or absence of ITB. The information that can be obtained is likely to be representative of the nature of the tumor at the time of diagnosis. Based on many studies, only an integrated analysis that includes iTB will be able to predict the patient’s prognosis more accurately. Furthermore, the clinicopathologic significance of ITBs can vary depending on how they are interpreted, so more detailed analysis may be required.

According to the ITBCC guidelines, TB is graded as low (<5 buds), intermediate (5–9 buds), or high (≥10 buds). Cases on the borderline between grades may require additional scrutiny. Various factors, including the selection of tissue section, evaluation area, and staining method (H&E or immunohistochemistry), can influence TB grading [1,8,15]. Observer variability is also a significant concern. Simple or straightforward criteria could reduce interobserver variability and enhance reproducibility. In our study, we evaluated ITB at a hotspot (area = 0.785 mm^2^), similar to the method for PTB. We identified the number of ITBs within the tumor and evaluated the clinicopathological significance of ITB in CRCs. Additionally, we investigated the correlation between ITB and TILs in CRCs. The ITBCC guidelines mention a correlation between ITB and lymph node metastasis. Our study found that high ITB was associated with aggressive tumor behavior, including lymph node metastasis, and significant correlations with lymphatic invasion, metastatic lymph node rate, tumor differentiation, vascular invasion, perineural invasion, and a higher pT stage. Marx et al. used a tissue microarray with a diameter of 0.6 mm, resulting in a high ITB rate of 32.2% [8]. We evaluated the entire section, not just a tissue microarray. Compared to the hotspot area (0.785 mm^2^), the 0.6 mm diameter represents a smaller area, which may explain their higher rate of high ITB, especially since they used pan-cytokeratin staining [8]. However, the ITBCC recommends evaluating TB using H&E [1]. Their findings on ITB’s association with tumor location, grade, pT stage, and lymphovascular invasion align with ours [8]. Some researchers have assessed ITB’s prognostic implications by counting in a high-power field or across 10 high-power fields, indicating a difference in the area between a hotspot and these fields [5,9,16,17]. This discrepancy could be due to the varying amounts of stroma in the two measurement methods, necessitating a comparison between these methods.

In our study, we focused on the prognostic implications of the intermediate grade (5–9 buds) within the 3-tier grading system for ITB. We analyzed survival rates according to ITB grades and observed significant prognostic stratification (*p* = 0.009). Notably, while the intermediate grade showed a significant difference in prognosis compared to the low grade, it did not significantly differ from the high grade. This finding suggests that the intermediate grade has a worse prognosis, similar to the high grade, supporting its inclusion in the high-ITB category in a simplified 2-tier system. To enhance reproducibility, we categorized patients into high- and low-ITB groups using a threshold of 5 TBs, and evaluated the clinicopathological significance of ITB. Previous studies have used various criteria for their 2-tier systems, with the number of ITBs ranging from 2 to 10 [16,18,19]. For instance, Lugli et al. defined high ITB as 7 or more buds, reporting a high ITB rate of 48.4% and correlating high ITB with aggressive tumor behavior [16]. In our study, the high ITB rate was 22.2%, and similar to these studies, we found that patients with high ITB had a worse prognosis than those with low ITB. Marx et al. categorized ITBs into three groups to assess their prognostic implications, revealing clear differences in survival among these groups [8]. However, they noted no significant difference in survival rates between the low- and intermediate-grade groups in patients with stage II CRC [8]. Given the significant results yielded by both the 2-tier and 3-tier systems, further research is necessary to compare these systems and potentially simplify the grading systems.

Numerous studies have investigated the role of TILs in patients with CRC. According to IS, CRCs can be classified into hot and cold tumors. Improving the efficacy of immunotherapy is associated with converting cold tumors into hot tumors or T-cell-inflammatory tumors [20]. In our previous work, we highlighted the significance of TILs [12]. We found that a low IS was associated with a poor prognosis. Interestingly, our data showed that the number of ITBs was higher in patients with high IS compared to those with low IS (3.28 ± 3.31 vs. 2.19 ± 2.59). Furthermore, we differentiated TILs into CD3-positive and CD8-positive lymphocytes to examine their correlation with ITB. A significant association was observed between ITB and CD8-positive lymphocytes, but not with CD3-positive lymphocytes. This aligns with previous research indicating that patients with CD8-positive T cells in rectal adenocarcinoma had a higher rate of complete or near-complete response [11]. Additionally, we set the criterion for high ITB at two in a hotspot area (0.785 mm^2^) [18]. While many factors contribute to predicting a patient’s prognosis, no single factor can provide a complete stratification. In our study, we attempted to stratify patient prognosis using both IS and ITB. In the high-IS group, ITB did not significantly impact prognosis. However, in the low-IS group, patients with high ITB had a worse prognosis compared to those with low ITB. This finding suggests that a low IS is indicative of poor prognosis, but our study also revealed heterogeneity within the low-IS group. Interestingly, while low ITB generally correlated with a good prognosis, we found this group to be heterogeneous as well. Therefore, ITB and IS alone may not be sufficient for comprehensive prognosis stratification in CRC. ITB is one aspect of tumor progression that can be seen, and it is known that this aspect can help predict prognosis. However, as our results reveal, there is such wide variation in presentation between patients that an integrated analysis will be required.

In our recent study, we conducted and reported a comprehensive meta-analysis focusing on the clinicopathological significance of ITB in CRC [2]. Our analysis was specifically focused on ITB, intentionally excluding PTB from our scope, which was included in previous analyses. This approach was taken to isolate and understand the unique implications of ITB. In our earlier findings, we observed that the low-TILs group had a high ITB rate of 25.0%. In contrast, the high ITB rate was 21.2% in the group with high TILs. However, there are limitations to the interpretation of the existing literature because the comparison is not based on the same criteria as IS. Furthermore, the results of the articles included in the meta-analysis were not evaluations of IS. They were evaluating peritumoral infiltrating lymphocytes and their association with ITB [8,9,16]. In this study, we evaluated the relevance of ITBs and TILs. In detail, we divided CD3- and CD8-positive TILs to assess their relevance. However, in our current cohort, we noted a significant difference in high ITB rates between the high- and low-IS groups, with rates of 11.1% and 29.7%, respectively. This indicates a notable variance in high ITB rates based on IS. Interestingly, our findings contrast with those of Ramadan et al. [9], who reported no significant correlation between TILs and ITB. Moreover, our study diverges from previous meta-analyses by attempting to stratify the prognosis of CRC patients using both ITB and IS as indicators. This approach aims to provide a more nuanced understanding of CRC prognosis by integrating these two key factors.

## 5. Conclusions

High ITB was frequently identified in 22.2% of overall CRC cases. High ITB was significantly associated with aggressive tumor characteristics, including lymphatic, vascular, perineural invasion, and lymph node metastasis. In addition, there was a significant correlation between high ITB and worse prognosis in patients with CRCs. ITB is also useful for determining the prognosis of CRCs with low IS. Therefore, using both ITB and IS for prognostic stratification can be an effective approach to predicting outcomes for CRC patients. Our results confirm the usefulness of the 2-tier system for evaluating ITB. Further large-scale studies may be needed before it can be used in daily practice.

## Figures and Tables

**Figure 1 jcm-13-00134-f001:**
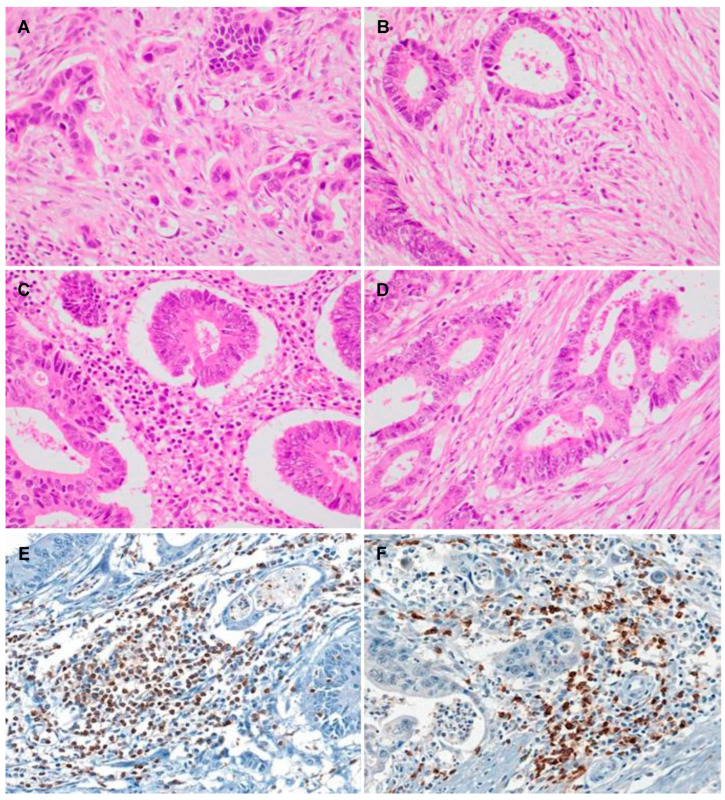
Representative images show colorectal cancers with intratumoral budding (ITB) (**A**–**D**). (**A**) Conventional colorectal adenocarcinoma with high ITB (×400). (**B**) Conventional colorectal adenocarcinoma with low ITB (×400). (**C**) Colorectal adenocarcinoma with high levels of tumor-infiltrating lymphocytes (×400). (**D**) Colorectal adenocarcinoma with low levels of tumor-infiltrating lymphocytes (×400). (**E**) CD3-positive lymphocytes (×400). (**F**) CD8-positive lymphocytes (×400).

**Figure 2 jcm-13-00134-f002:**
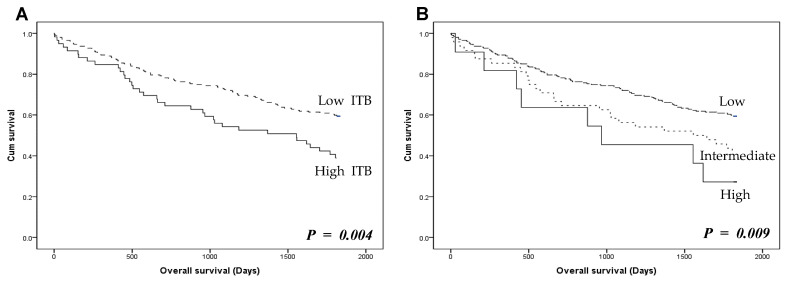
Overall survival according to the presence of intratumoral budding (ITB) in colorectal cancers. (**A**) Overall survival of 2-tier ITB grading system. High ITB shows worse overall survival than low ITB (log-rank, *p* = 0.004) (**B**) 3-tier ITB grading system is associated with overall survival (log-rank, *p* = 0.009). *p* < 0.05 are highlighted in italic bold.

**Figure 3 jcm-13-00134-f003:**
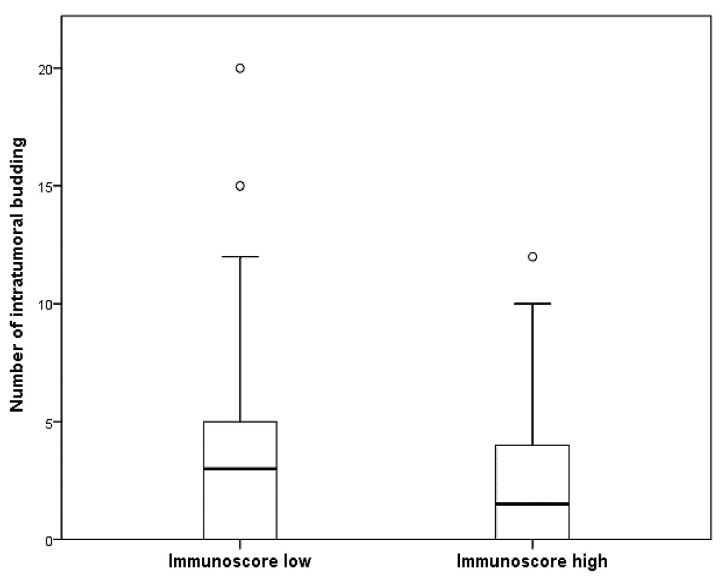
Comparison of the number of intratumoral buddings (ITBs) between high- and low-immunoscore (IS) groups in colorectal cancers. The number of ITB is 3.28 ± 3.31 for low-IS group and 2.19 ± 2.59 for high-IS group, with a significant difference (*p* = 0.005).

**Figure 4 jcm-13-00134-f004:**
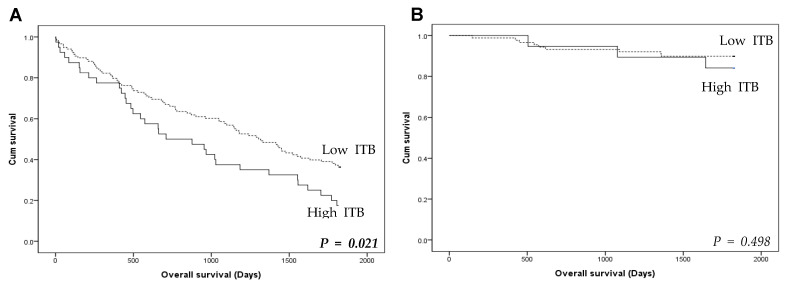
Stratification of overall survival according to intratumoral budding (ITB) and immunoscore (IS) in colorectal cancers. (**A**) Overall survival of low IS group. High ITB shows worse overall survival than low ITB in low-IS group (Log-rank, *p* = 0.021). (**B**) Overall survival of high IS group. No significant difference in overall survival between high and low ITBs (log-rank, *p* = 0.498). *p* < 0.05 are highlighted in italic bold.

**Table 1 jcm-13-00134-t001:** The correlation between intratumoral budding and clinicopathological parameters in colorectal cancers.

	Intratumoral Budding		*p*-Value
	High	Low	
Total (*n* = 266)	59 (22.2)	207 (77.8)	
Age (years)	65.24 ± 12.60	63.12 ± 12.99	0.266
Sex			
Male	30 (50.8)	105 (50.7)	1.000
Female	29 (49.2)	102 (49.3)	
Tumor size			
≤5 cm	30 (50.8)	76 (36.7)	0.070
>5 cm	29 (49.2)	131 (63.3)	
Tumor size (cm)	5.04 ± 1.82	5.58 ± 2.13	0.074
Location of tumor			
Right colon	33 (55.9)	95 (45.9)	0.186
Left colon	26 (44.1)	112 (54.1)	
Tumor differentiation			
Well or Moderately	41 (69.5)	170 (82.1)	** *0.045* **
Poorly	18 (30.5)	37 (17.9)	
Vascular invasion			
Present	10 (16.9)	14 (6.8)	** *0.035* **
Absent	49 (83.1)	193 (93.2)	
Lymphatic invasion			
Present	30 (50.8)	40 (19.3)	** *<0.001* **
Absent	29 (49.2)	167 (80.7)	
Perineural invasion			
Present	21 (35.6)	23 (11.1)	** *<0.001* **
Absent	38 (64.4)	184 (88.9)	
pT stage			
pT1–2	4 (6.8)	37 (17.9)	** *0.041* **
pT3–4	55 (93.2)	170 (82.1)	
Lymph node metastasis			
Present	40 (67.8)	106 (51.2)	** *0.026* **
Absent	19 (32.2)	101 (48.8)	
Metastatic lymph node ratio	0.21 ± 0.27	0.11 ± 0.20	** *0.008* **
Distant metastasis			
Present	11 (18.6)	18 (8.7)	0.055
Absent	48 (81.4)	189 (91.3)	
pTNM stage			
I–II	30 (50.8)	80 (51.6)	0.055
III–IV	29 (49.2)	75 (48.4)	

Numbers in parentheses represent percentages. *p* < 0.05 are highlighted in italic bold.

**Table 2 jcm-13-00134-t002:** The correlation between intratumoral budding and tumor stromal parameter in colorectal cancers.

	Intratumoral Budding		*p*-Value
	High	Low	
Total (*n* = 266)	59 (22.2)	207 (77.8)	
Immunoscore			**<*0.001***
High	12 (20.3)	96 (46.4)	
Low	47 (79.7)	111 (53.6)	
CD3-positive lymphocytes			0.283
High	17 (28.8)	76 (36.7)	
Low	42 (71.2)	131 (63.3)	
CD8-positive lymphocytes			** *0.044* **
High	14 (23.7)	80 (38.6)	
Low	45 (76.3)	127 (61.4)	

Numbers in parentheses represent percentages. *p* < 0.05 are highlighted in Italic bold.

## Data Availability

Data are contained within the article.

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
