# Peer review of "Prognostic Implications of Intratumoral Budding in Colorectal Cancer: Detailed Analysis Based on Tumor-Infiltrating Lymphocytes"

_jcm, 2023, doi:10.3390/jcm13010134_

Round 1
Reviewer 1 Report
Comments and Suggestions for Authors
The article concerns a relatively new research topic which is tumor budding. The know;edge about this subject has been gradually increasing over the past few years. There are only a handful of studies in the CRC field.
The introduction provides a sufficient background for the topic. The methods are clear and accurate.
The results section is brisk and contains interesting findings which are likely to become a cornerstone for future studies.
The conclusions are fair and supported by the results.
The number of refferences is somewhat low. It'd be advantageous if more were included.
Author Response
We tried to address the points raised by the reviewers as best as we could. The specific responses to the reviewers’ comments are described.

Reviewer 2 Report
Comments and Suggestions for Authors
Review Report
Manuscript ID: jcm-2756728
Title: Prognostic implications of intratumoral budding in colorectal cancer: Detailed analysis based on tumor-infiltrating lymphocytes
To predict the prognosis of CRC patients and present references to CRC clinical treatment, the authors invested and divided CRC patients as two groups with high or low intratumoral budding (ITB), and analyzed their correlation with tumor-infiltrating lymphocytes (TILs) and immunoscore (IS). This is a novel investigation to enhance the diagnosis and clinical pathological evaluation of CRC patients by a new approach.
Comments:
1. The English writing for this manuscript is not so good and should be revised by a native English person. Such as in ABSTRACT, “The clinicopathological significance of ITB in human CRCs was evaluated, and a detailed analysis based on tumor-infiltrating lymphocytes (TILs) was also performed”, “human CRCs” and “performed” were described not so English and professionally, and “tumor-infiltrating lymphocytes” and its abbreviation appeared twice!
2. It is poor to validate the reality of the manuscript because only one microscopy is included in it. The authors can add the photos about the intensity of duding with vascular forming, classified display of different TILs by immunohistochemistry or immunohistofluorescence.
3. In the ABSTRACT and DISCUSSION, the authors must use the concepts of hot tumor and cold tumor.
Overall Opinion and Suggestion: It is an important and useful manuscript, but it needs to be largely revised!
Comments on the Quality of English LanguageReview Report
Manuscript ID: jcm-2756728
Title: Prognostic implications of intratumoral budding in colorectal cancer: Detailed analysis based on tumor-infiltrating lymphocytes
To predict the prognosis of CRC patients and present references to CRC clinical treatment, the authors invested and divided CRC patients as two groups with high or low intratumoral budding (ITB), and analyzed their correlation with tumor-infiltrating lymphocytes (TILs) and immunoscore (IS). This is a novel investigation to enhance the diagnosis and clinical pathological evaluation of CRC patients by a new approach.
Comments:
1. The English writing for this manuscript is not so good and should be revised by a native English person. Such as in ABSTRACT, “The clinicopathological significance of ITB in human CRCs was evaluated, and a detailed analysis based on tumor-infiltrating lymphocytes (TILs) was also performed”, “human CRCs” and “performed” were described not so English and professionally, and “tumor-infiltrating lymphocytes” and its abbreviation appeared twice!
2. It is poor to validate the reality of the manuscript because only one microscopy is included in it. The authors can add the photos about the intensity of duding with vascular forming, classified display of different TILs by immunohistochemistry or immunohistofluorescence.
3. In the ABSTRACT and DISCUSSION, the authors must use the concepts of hot tumor and cold tumor.
Overall Opinion and Suggestion: It is an important and useful manuscript, but it needs to be largely revised!
Author Response

(The authors gave the same response as above.)

Reviewer 3 Report
Comments and Suggestions for Authors
This is a retrospective study about the prognostic implications of tumor-infiltrating lymphocytes in colorectal cancer. A lot of literature is available on this theme.
The novelty of this study is to attempt to elucidate the correlation between intratumoral budding and tumor-infiltrating lymphocytes in colorectal cancer. Additionally, the prognostic implications of ITB were evaluated and stratified using TILs.
Although this is a retrospective article, the methods are adequately described and the results are clearly presented.
The plagiarism is found in introduction. Please correct.
Author Response

(The authors gave the same response as above.)

Round 2
Reviewer 2 Report
Comments and Suggestions for Authors
Overall Opinion and Suggestion:
Although the authors added the photo of immunohistochemistry TILs in Fig 1, the intensity of budding with vascular forming is not validated well with only one figure.
But anyhow, I think it is more acceptable than original manuscript.
Comments on the Quality of English LanguageAcceptale.